# A Novel RUL Prognosis Model Based on Counterpropagating Learning Approach

**Mohammed Baz** 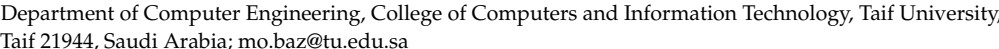

Department of Computer Engineering, College of Computers and Information Technology, Taif University, Taif 21944, Saudi Arabia; mo.baz@tu.edu.sa

**Abstract:** The aviation industry is one of the fastest-growing sectors and is crucial for both passenger transport and logistics. However, the high costs associated with maintenance, refurbishment, and overhaul (MRO) constitute one of the biggest challenges facing this industry. Motivated by the significant role that remaining useful life (RUL) prognostics can play in optimising MRO operations and saving lives, this paper proposes a novel data-driven RUL prognosis model based on counter propagation network principles. The proposed model introduces the recursive growing hierarchical self-organisation map (ReGHSOM) as a variant of SOM that can cluster multivariate time series with high correlations and hierarchical dependencies typically found in RUL datasets. Moreover, ReGHSOM is designed to allow this clustering to evolve dynamically at runtime without imposing constraints or prior assumptions on the hypothesis spaces of the architectures. The output of ReGHSOM is fed into the supervised learning layers of Grossberg to make the RUL prediction. The performance of the proposed model is comprehensively evaluated by measuring its learnability, evolution, and comparison with related work using standard statistical metrics. The results of this evaluation show that the model can achieve an average mean square error of 5.24 and an average score of 293 for the C-MPASS dataset, which are better results than most of the comparable works.

**Keywords:** counter propagation neural network; self-organising map; recursive SOM

## 1. Introduction

In the current decade, the use of aerospace technology in passenger transport and logistics has increased significantly. According to [1], the global market for aerospace forgings was estimated at USD 30 billion by 2022, and this figure is expected to rise to more than USD 50 billion by the end of 2035, with a compound annual growth rate of 8%. However, one of the biggest challenges facing this industry is the high cost of maintenance, refurbishment, and overhaul (MRO). According to [2], MRO costs were estimated at USD 62 billion in 2021, representing approximately 11.2% of total airline operating costs, with engines being the largest cost segment at 37% of these costs.

Remaining useful life (RUL) prognostics is one of the very effective strategies widely used to optimise MRO operations. The main objective of prognostics is to accurately predict how long an asset can continue to perform its intended function [3,4]. Such prediction enables an MRO operation to be performed in accordance with the actual condition of the component, which, in turn, can save costs on unnecessary MRO operations. Indeed, several studies, e.g., [3,4], report that the high fluctuation of usage patterns and operating conditions makes the scheduled maintenance inaccurate, while [5] estimates that a total of USD 3 billion is wasted on the no fault found (NFF) inventory. In addition to the direct monetary savings that can be achieved through prognostics, a reduction in MRO activities can also reduce the human errors that occur during this process. Such a reduction is not insignificant but accounts for about 80% of MRO errors [6,7]. Another key advantage of prognosis-based MRO optimisation is its ability to extend component life and reduce maintenance delay [8,9]. A study conducted by the authors of [8] on a fleet of 100 long-haul

air-craft engines shows that prognosis-based MRO can prolong engine life by about 30–40%, while [9] shows that a 20% reduction in maintenance time can be achieved. Besides the above benefits, some studies show the benefits of using prognostics in terms of improving spare parts supply chains [10], increasing fleet availability [11], and reducing collateral damage during repairs and on-ground aviation [12].

The significant benefits that RUL prognosis can offer to the aviation sector have sparked the interest of researchers to use the most advanced modelling approaches to optimise MRO operations. Many of the pioneering models use analytical approaches to develop mathematical models capable of characterising the degradation behaviour of physical systems, e.g., [13–18]. Although these models demonstrate their feasibility, the need to account for various interactions within the modelled system and its operating conditions can pose challenges to their solvability and in some cases make them intractable [19]. In response to these challenges, the principles of data-driven models have been utilised as contemporary modelling techniques. Data-driven models use external observations generated by the system to make predictions about its degradation status. This, in turn, makes data-driven models well suited to the continuous evolution that modern aviation systems undergo [20]. Among the various approaches used in data-driven models, the deep artificial neural network (DANN) has emerged as the mainstream architecture. Simply put, a DANN [21] is an acyclic graph consisting of computational units with learnable parameters organised into layers and connected by an objective function. During training, the model is presented with a set of observations along with their corresponding desired outcomes. The model then attempts to adjust its learnable parameters so that its output is as close as possible to the desired outcomes. This adjustment process is usually facilitated by the backpropagation algorithm, which is an error correction learning strategy. In backpropagation, the errors calculated in the output layer are passed backwards to the preceding layers so that each layer adjusts its learnable parameters in response to the errors it produces. While several works using DANNs have achieved remarkable results by tailoring the hypothesis spaces of the model to the dataset, e.g., [22–37], it is important to recognise that there are several reported limitations associated with the DANN architecture. One of the most important limitations is the high vulnerability of DANN models to data and concept drift [38,39], the increase in gradient instabilities with the depth of the DANN model [21,40], the high sensitivity of DANN to noise and outlines in the datasets [21,41], and the high computational cost.

Motivated by the importance of developing a reliable data-driven prognosis model and the shortcomings of existing DANN models, this paper proposes a novel model based on the counter-propagation network (CPN) [42]. The core approach underlying the CPN is that a combination of unsupervised and supervised learning strategies within the same architecture can improve the learning capability of the model and solve some of the main problems related to backpropagation. In a CPN, a raw dataset is fed into a self-organisation map (SOM), which is a nonlinear vector quantisation algorithm known for its ability to preserve the topological order of its inputs [43]. The output of the SOM is then fed into a supervised network based on Grossberg's learning approach, which is known for its high fidelity and low computational budget. Several works comparing the performance of CPN with that of DANN, e.g., [44–49], show the superiority of CPN in terms of higher convergence speed, greater resilience to noise and outliers, better computational efficiency, better interpretability and explainability, and lower sensitivity to concept/data drifts.

However, one of the biggest challenges standing in the way of applying CPN directly to RUL prognosis tasks is the fact that the original CPN was designed to process time-agnostic datasets. In contrast, most datasets that characterise RUL are a collection of multivariate time series. Typically, each instance of such a dataset consists of a series of observations taken from various subcomponents of the assets being monitored. Due to the mutual coupling amongst these subcomponents and the fact that they are exposed to the same operating conditions, the RUL dataset usually exhibits a high degree of correlation and hierarchical dependencies. This work addresses the unique nature of RUL datasets

by proposing a novel variant of a SOM map dubbed recursive growing hierarchical SOM (ReGHSOM). ReGHSOM combines the strengths of ReSOM [50], which was developed to allow traditional SOMs to handle temporal relationships of the dataset through a fixed architecture, and GHSOM [51], which was developed to allow dynamic evolution of the SOM without considering temporal dependencies. This combination enables the proposed ReGHSOM algorithm to effectively handle high correlations and hierarchical dependencies of multivariate time series datasets. Indeed, ReGHSOM does not impose any constraints or prior assumptions on the architectures of the model, which, in turn, allows ReSOM to deal with different shapes of datasets without having to seek the suitability of the model's hypothesis spaces for the particular dataset. Another important aspect of the ReGHSOM is its ability to transform nonlinear statistical relationships embedded in multivariate time series observations into a simpler geometric representation that can preserve their topological order. Therefore, latent relationships can be thoroughly visualised and rigorously quantified. Furthermore, feeding the supervised layer with a meaningful and low-dimensional representation of the original dataset not only improves prediction accuracy by reducing the impact of noisy data points, but also enables a reduction in the time required for these predictions by reducing the computational complexity of the model.

The performance of the proposed model was comprehensively evaluated using the commercial modular aero-propulsion system simulation (C-MPASS) dataset [52]. This dataset was selected for evaluation because it is one of the most commonly used benchmarking datasets in multimodal work, allowing a fair comparison between the results of the proposed model and others. Another important feature of this dataset is that it uses different conditions, fault models, and noise levels to generate the readings. Performing the evaluations under these cases allows us to assess the suitability of the proposed model for dealing with quasi-real datasets. In addition to comparing the results of the model with relevant work, the evaluation of the proposed model also includes its learning ability and its evolution under different subsets of C-MPASS. All evaluations are conducted using standard statistical metrics, including mean absolute error, root mean square error, and score. The results of this evaluation show that the proposed model is able to achieve an average mean square error of 5.24 and an average score of 293 for the C-MPASS dataset, which are better than most of the comparable works.

To summarise, the main contribution of this work is to develop a versatile RUL prognostics model that can dynamically adapt its architecture to the characteristics of the degradation dataset in real-time. This adaptability extends the applicability of the model to entire engines or even specific components without requiring extensive adjustments to the model's hypothesis spaces. The high prediction accuracy that the proposed model can achieve makes it one of the valuable methods not only in optimising standard MRO operations but also in contemporary non-destructive testing (NDT) from different perspectives. This includes reducing the cost and efforts associated with performing unnecessary NDTs or MROs, as the prediction generated by the model can reveal the status of the system under its actual operational conditions. Moreover, incorporating the readings from different components in the proposed model facilitates predicting the performance of those components that cannot be easily inspected by the NDTs. Another important benefit of the proposed model is this context stems from its high computational feasibility, which, in turn, facilitates incorporating it with other operational processes seamlessly.

The rest of this paper is organised as follows: Section 2 reviews the most pertinent works presented in the open literature, Section 3 describes the proposed model, Section 4 presents the results and discussion, and Section 5 concludes this paper.

## 2. Related Works

The development of an effective model that can predict RUL or other related aircraft component degradation metrics is one of the most active research areas that has received much attention due to its key role in saving lives and optimising aviation MRO practices.

Most of the work presented in the open literature can be divided into two border groups: physics-based models and data-driven models.

The core concept underlying most of the physical-based models is that the behaviour exhibited by a system during its life cycle can be quantified mathematically. Therefore, the signs of deterioration can be identified simply by interpreting these models in the light of fundamental laws of science and their derivations [19]. Broadly speaking, most of these models can vary according to several criteria. These include the factors that contribute to degradation (e.g., environmental conditions (e.g., [13]) and operating conditions (e.g., [14])); the mechanisms by which degradation occurs (e.g., competitive degradation (e.g., [15]) and multistage degradation (e.g., [16])); and the methods used to represent uncertainty in the model (e.g., deterministic (e.g., [17]) or stochastic approaches (e.g., [18])). Although physical-based models have the potential to achieve a high level of fidelity, they often entail a significant trade-off between the level of details that go into the models and the solvability of the model. A very detailed model can represent the complexity of the real world but may be difficult to solve, while a simplified model may be easier to work with but may not fully represent real-world scenarios. Another notable limitation of physical-based models is their lack of versatility, as models developed for specific machines or systems cannot be easily applied to other machines.

Data-driven models, on the other hand, are based on the assumption that the degradation characteristics of a system can be determined by analysing the observations generated by that system. This, in turn, makes data-driven models advantageous as they do not require tracking the internal state space of systems or a mathematical representation of the machine. The proliferation of high-precision sensors and rapid advances in the field of deep learning model artificial intelligence further reinforce this trend by facilitating the integration of extensive sensor-derived information for accurate predictions.

Deep artificial neuron networks (DANN) are one of the predominant modelling approaches in this context. The work presented in [22] proposes a deep learning model for predicting the remaining useful life of aircraft turbofan engines. In developing this model, it was assumed that removing outliners and noisy data points can reduce the time and computational complexity of the model, which, in turn, can lead to a faster learning curve and better prediction readings. Therefore, four preprocessing phases were applied to the raw dataset. In the first phase, a correlation analysis is performed between the RUL values and the sensor trajectories for each sub-dataset. All trajectories whose correlation coefficients are less than 10% are excluded from the subsequent preprocessing phases, while the remaining trajectories are run through a moving median filter with an adaptive time window. In the third and fourth preprocessing phases, Z-score normalisation and an improved piecewise linear degradation model are used. The proposed model uses the LSTM, drop-out, and fully connected architectures to obtain the RUL values, while the iterative grid search technique is used to adjust the hyperparameters of the model (including the number of layers, the number of neurons in each layer, batch size, etc.). In this work, the C-MAPSS dataset is used to evaluate the accuracy of the proposed models. The results of this evaluation show that the prediction metrics vary between the sub-datasets, with the highest achievable root mean square error ($RMSE$) being 7.78, while the lowest is 17.63. The work presented in [23] follows the same procedures presented in [22], but uses different preprocessing techniques. More specifically, this work uses maximum information coefficient theory (MICT) instead of the correlation analysis used in [22] to determine the degree of association between sensor trajectories and the given RUL in each training subset of C-MAPSS. The processed data are then treated with a technique that combines both the simple moving average method and kernel principal component analysis to smooth the noisy data points and map the remaining data points to a low-dimensional space before feeding them into the deep learning model. The proposed model consists of a series of LSTM layers followed by drop-out and fully connected layers. The results of this work show that the highest RSME value is 9.65 and the worst is 22.21. Following the same modelling approach, the authors of [24] investigate the impact of different correlation-based

filtering methods and feature selection wrapper techniques on the prediction performance using the C-MAPSS benchmark dataset. In this work, the MLP architecture with different number of layers and neurons is used. The results of this work show that the best *RMSE* value of 44.71 can be achieved when using the evolutionary wrapper selection method with four fully connected layers followed by a drop-out layer and a single layer.

The great success that the convolutional neural network (CNN) has achieved in computer vision and related disciplines has inspired a cadre of scholars to use it to predict RUL. A CNN-based model relies mainly on the ability of this network architecture to extract the salient feature automatically without a need for pre-adjusting. The work presented in [25], for example, proposed a CNN model consisting of two pairs of convolutional layers, each followed by a pooling layer and a fully connected layer from which the predicted RUL values are derived. A sliding window of length 15 is used to segment the multivariate time series of the raw datasets into smaller units before processing them with the proposed CNN model. The results of this work show that the proposed model performs better compared to the other three models developed using MLP, support vector regression (SVR), and relevance vector regression (RVR). However, the highest *RMSE* value of 18.4480 reported by the proposed model was not higher than the values reported by comparable works. The work in [26] is another example that uses a CNN architecture to predict the RUL values of C-MAPSS data. This work aims to reduce the loss of information that results from the change in dimensionality of the dataset when it is processed through convolutional layers. The idea of this work is to use zero-padding convolutional layers for primitive feature extraction and a unit kernel convolutional layer for combining all previously extracted features. Despite the tolerable performance values of the proposed model, the comprehensive evaluation of CNN architectures in the context of RUL prediction concludes that there is a proportional relationship between the number of convolutional layers and the prediction performed, but this advantage is outweighed by the computational budgets and training time.

Besides the above, there are other works that aim to improve predictive performance by integrating different ANN architectures over the same model, using continual learning techniques and federated learning principles. An example of this direction is [27], which uses three different architectures: (i) CNN to extract the features from the dataset, (ii) convolution block attention module to discriminate the most relevant features and discard the rest of the features extracted by CNN, and (iii) LSTM to reveal the latent relationships between selected features and the predicted RUL. The result of this work shows the ability of the proposed model to achieve an *RMSE* of 5.50 on the C-MPASS dataset, but there is no further information about if this value is due to the whole dataset or just a part of it. An example of the use of continual learning was presented in [28]. The basic idea of this work is to use the elastic weight consolidation (EWC) approach to mitigate the negative impact of catastrophic forgetting on prediction performance. Catastrophic forgetting is one of the well-known limitations of deep learning models. It occurs when the model cannot retrieve the knowledge it gained from processing previous samples when a more recent instance is processed. EWC addresses this limitation by regulating the model's parameter spaces according to the importance of the acquired knowledge. The performance reported in this paper shows that it outperforms other models based on CNN and restricted Boltzmann machine and LSTM architectures. The authors of [29] propose a federated learning model where the learning tasks are distributed across multiple nodes rather than the exhaustion of a single machine resource by a massive training dataset. The performance evaluation of the proposed model is performed using both weight aggregation algorithms, synchronous and asynchronous, and it is shown that a higher value can be obtained with the proposed model. The work proposed in [30] provides a new perspective on the development of RUL estimation models by assuming that this estimation can be formulated as a decision-making problem rather than a regression problem, as is the case in other work. In this work, the Markov decision process is used to model the set of observations in the dataset as a linked state space, while deep reinforcement learning is used as a means to identify the best estimation strategy. The work proposed in [31] attempts to

overcome the high complexity of traditional spatio-temporal deep learning by proposing a lightweight operator and using it with the GRU architecture. In this work, it is claimed that the proposed operator is able to extract the relevant information for the given dataset and seamlessly insert it into the following layers of the model. In addition, some recent works such as [32–37] have been devoted to improving the prediction performance by incorporating one or more of the above architectures.

## 3. The Proposed Model

The main contribution of this work is to develop a novel data-driven model that can predict the values of RUL based on counter propagation network (CPN) principles [42]. This approach was chosen for its robustness in processing large amounts of multivariate data, even when contaminated with noise and outliers. In addition, CPN is known for its effective learning ability and fast convergence. Because of these properties, CPN has been used to solve real-world problems with intricate data structures, including the mapping and interpretation of infrared spectra of compounds [53], inferring the molecules behave in acidic or basic environments [54], phylogenetic classification of ribosomal RNA [55], structural analysis and design [44].

The CPN framework was introduced by Hecht-Nielsen as a hybrid artificial neural network that seamlessly integrates supervised and unsupervised learning strategies into a single architecture. In the unsupervised learning phase, the self-organising map (SOM) [43] is used to encode the high-dimensional input data into a low-dimensional space that preserves its topological order, while in the supervised learning phase, the Grossberg network is used to associate the low-dimensional representations generated by the SOM to a set of target outcomes. CPNs can be constructed in two main configurations: a full CPN and a forward-only CPN. A full CPN consists of two input layers, a SOM map, and two Grossberg layers. The two input layers are designed to receive a set of observations and the corresponding target outputs, while the two Grossberg layers are responsible for generating the best possible approximations of these inputs. The SOM layer acts as a mediator that facilitates the transformation between the output and input spaces. In contrast, in a forward CPN, the readings are received by an SOM layer and an approximation is generated by the Grossberg layer. This makes the full CPN suitable for bidirectional function approximation and the forward-only CPN suitable for unidirectional function approximation. Considering that the main objective of the proposed model is to map the set of observations into RUL values, the suitability of the forward-only CPN for this purpose becomes clear. However, since the original forward-only CPN architecture was designed for processing non-sequential datasets, a new form of this architecture is proposed here. In our proposal, the recursive SOM [50] is combined with the growing hierarchical SOM [51] architecture to form a novel unsupervised learning model, which we refer to here as recursive growing hierarchical SOM (ReGHSOM), which effectively processes RUL data. To illustrate the proposed model, Figure 1 shows a high-level abstraction of the different components of this model. As you can see, the multivariate time series of sensor readings are fed into the unsupervised layer (ReGHSOM), which clusters them hierarchically to reflect the different granularity of the dataset. The centre of each cluster, represented in colours (also known as best matching units), is connected to the supervised layer (Grossberg), from which the predicted RUL values are generated. The rest of this section is organised as follows: Section 3.1 provides a formal description of the RUL prediction and the underlying assumptions used to develop the model. Sections 3.2 and 3.3 provide a detailed description for the ReGHSOM and Grossberg layer, respectively.

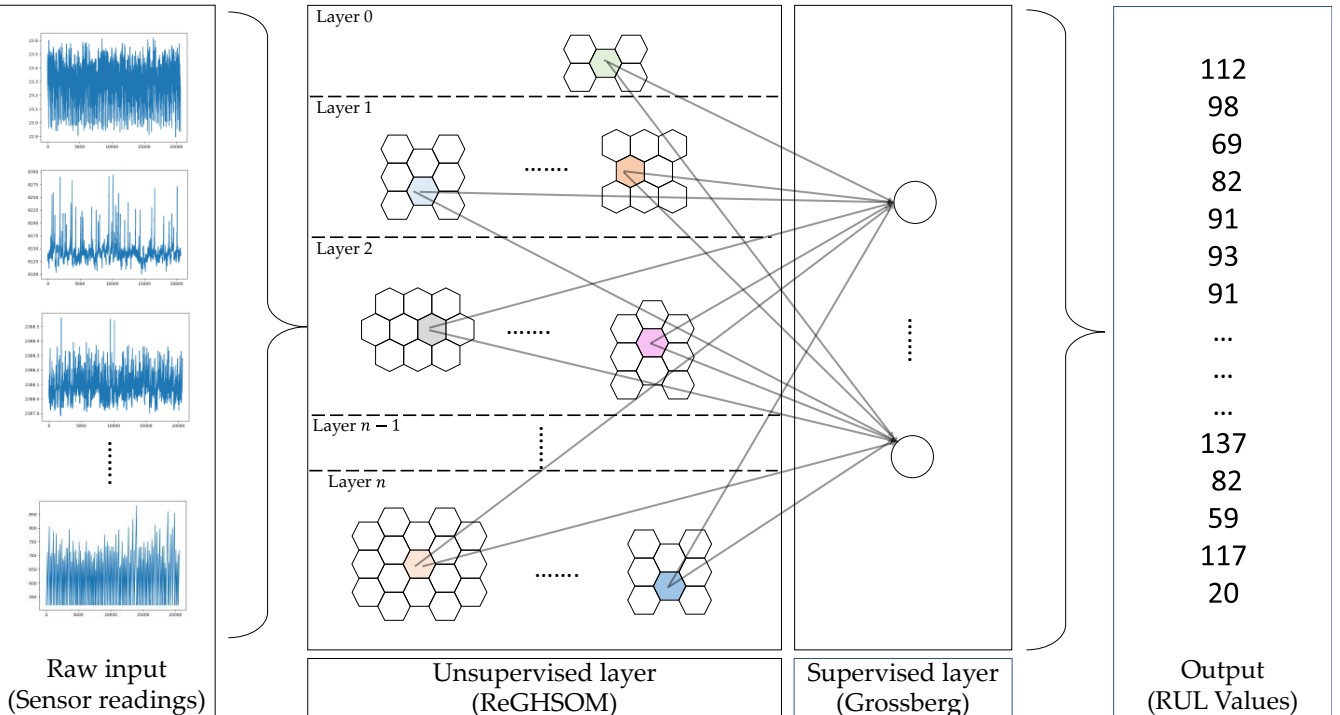

**Figure 1.** High level abstraction of the proposed model.

### 3.1. Problem Formulation and Underlying Assumptions

This study considers a collection of multivariant time series, denoted by $\mathcal{A}$, representing measurements of different parts of several assets and multiple conditions under which these assets are operate;, the term observations is used here to refer to these measurements and operating conditions collectively. The dimensionality of $\mathcal{A}$ is defined as $\mathcal{A} \in \mathbb{R}^{N \times \|\mathcal{E}\| \times \Sigma_{\forall e \in \mathcal{E}} t_e}$ where $N$ is the width of observations, $\mathcal{E}$ is a set containing identifiers of all assets accommodated $\mathcal{A}$, $\|\cdot\|$ is the cardinality symbol, and $t_e$ is the number of cycles at which the observations related to arbitrary assets are is monitored. For the sake of generality, it is assumed that the monitoring cycles of different assets are not necessarily congruent, i.e., $t_e \not\equiv t_d; \forall e, d \in \mathcal{E}$. Furthermore let $\mathcal{A}_e$ be the subset of $\mathcal{A}$ that contains all observations related to the engine e, $\mathcal{A}_e \subseteq \mathcal{A}$ this set can be written as $\mathcal{A}_e = \{a_{e,n}(t)\}_{t=1}^{t_e} = \{a_{e,1}(t), a_{e,2}(t), \ldots, a_{e,N}(t_e)\}_{t=1}^{t_e}$, and $\mathcal{R}$ be the set of length $\|\mathcal{E}\|$ containing the RUL values of all assets. Based on the above, the goal of a data-driven model is to find such a function $f$ that accepts the set of observations $\mathcal{A}$ as input and produces a vector of values that are as close as possible to the real RUL values, denoted here by $\hat{\mathcal{R}}$; hence, $f$ can be expressed as $f : \mathcal{A} \rightarrow \hat{\mathcal{R}}$.

A data-driven model aims to derive $f$ by applying a learning strategy to a collection of learnable computing units constructed according to a particular hypothesis space. Although there are no golden rules that can be followed in defining the hypothesis space or the learning strategy, here we attempt to discuss the underlying assumptions used in developing the proposed model. First, the use of a type of computing unit that can recognise the temporal structure embedded in time series, which is strongly required due to the fact that most of the observations provided by RUL datasets are time series. Second, the use of a versatile hypothesis space that can be easily adapted to the structure and complexity of the dataset.

### 3.2. RGHSOM Unsupervised Layer

The self-organising map (SOM) is a type of connectionist system introduced by Kohonen in 1982 [43] and is, therefore, also referred to as the Kohonen map in some references. SOM was inspired by the mechanism by which cortical maps evolve automatically during

growth. Indeed, several research studies on neuronal information processing have shown that interactions between cortex cells in response to a given stimulus are dominated by their lateral spacing. In this process, cells that are better able to interpret a stimulus increase their activation by emitting excitatory signals to their neighbours, while keeping distant cells in suspension by sending inhibitory signals. These interactions lead to self-organisation of cortical maps in a topographically meaningful order.

SOM resembles the self-organising phenomena of the brain described above in that it combines the competitive learning approach [56,57] with a spatiotemporal function called the neighbourhood function. In its simplest form, a SOM consists of several artificial neurons arranged in a two-dimensional lattice. Each neuron in the map is connected to all neurons in the input layer by a weighting vector, referred to here as the receptive weighting vector (it is also called the codebook vector or prototype), whose dimension is set according to the number of neurons in the input layer. In addition, each neuron in the SOM map is connected to the other neurons in the same layer by either an excitatory or an inhibitory weighting, depending on their lateral distance. During the training phase, all weighting vectors are randomly initialised, after which an instance of the observation dataset is presented to the SOM. The neuron in the SOM map then applies some kind of radial function to calculate the extent to which its receptive weighting vector matches the presented instance. The neuron with the best match is then nominated and this begins the weighting update process, in which the receptive weighting vectors of the unit with the best match and its neighbours are moved closer to the given readings, while the vectors of the other neurons remain unaffected. At the end of the training phase, the SOM should be able to transform nonlinear statistical relationships embedded in high-dimensional observations into a simpler geometric representation that can preserve their topological order.

However, the lack of effective mechanisms by which the standard SOMs can incorporate temporal dependencies into their clustering formation, as well as their rigid topologies, stand in the way of straightforward application of SOMs to RUL prediction. Some works have focused on improving SOM capabilities for processing sequential datasets (e.g., temporal SOM, hypermap, recurrent SOM, and recursive SOM) [58], while others have concentrated on extending the SOM topology according to the nature of the dataset under consideration (e.g., growing SOM [59] and growing hierarchical SOM [51]). This work aims to enhance the capabilities of SOM in both perspectives by combining ReSOM with GHSOM. The underlying approach on which ReSOM was developed is to allow the classical SOM to learn from its past activities by feeding it with a lagged-in-time copy of the SOM as additional input. Therefore, at a time instant, the neurons of a ReSOM receive two homogeneous inputs: the first is a feedforward input representing the instances of the training dataset corresponding to that time point, and the second input is the activity of the SOM generated at the delayed time step. These two inputs are concatenated and then fed into a classical SOM map. This, in turn, allows the ReSOM to follow the same procedures and mechanisms of the classical SOM map, including learning rules, weight updates, and neighbourhood function.

The principle of the GHSOM, on the other hand, is to construct a SOM map that can grow dynamically in accordance with the dataset that the map encounters at runtime. Such growth can occur vertically by adding new maps to the existing structure and horizontally by adding new neurons to the same map. This process continues until a suitable SOM topology emerges that can effectively represent the different patterns exhibited by the datasets and their relationships to each other.

To explain how the integration of ReSOM and GHSOM works, we assume that at a time instant $k$ there is a SOM map with a number of neurons at level $j$; here, $\boldsymbol{M}$ is used to refer to the set that accommodates these neurons. The feedforward and feedback weight vectors of any neuron $m_j$ at time $k$, i.e., $m_j \in \boldsymbol{M}$, are denoted by $\boldsymbol{w}_{m_j}^x[k]$ and $\boldsymbol{w}_{m_j}^y[k]$, respectively, where the $\boldsymbol{w}_{m_j}^x[k] \in \mathbb{R}^N$ and $\boldsymbol{w}_{m_j}^y[k] \in \mathbb{R}^{\|\mathbf{M}\|}$. At this point, the map is presented

with a realisation of the input, denoted $r(t) \subseteq \mathcal{A}$, and each neuron calculates its distance with respect to $r(t)$ as:

$$D_{m_j}[k] = -\alpha \left( \left\| r(t) - w^x_{m_j}[k] \right\| \right)^2 - \beta \left( \left\| y_{m_j}[k-1] - w^y_{m_j}[k] \right\| \right)^2 \tag{1}$$

where $\alpha$ and $\beta$ are hypermeters used to control the how long the historical information is involved in the computing the distance, whereas $y_{m_j}[k-1]$ is the output generated by this neuron at the preceding time step which is computed as:

$$y_{m_j}[k-1] = \exp\left( D_{m_j}[k-1] \right) \tag{2}$$

The neuron with the minimum distance to the given input at time instance $k$ is nominated as best matching unit (BMU), i.e., $m_j^* = \arg \min_{m_j} \left\{ D_{m_j}[k] \right\}_{\forall m_j \in M}$ and then weight vectors are updated according to:

$$w^x_{m_j}[k+1] = w^x_{m_j}[k] + \gamma[k] h_{m_j, m_j^*} \left( r(t) - w^x_{m_j}[k] \right)$$
$$w^y_{m_j}[k+1] = w^y_{m_j}[k] + \gamma[k] h_{m_j, m_j^*} \left( y_{m_j}[k-1] - w^y_{m_j}[k] \right) \tag{3}$$

where $\gamma[k]$ is the learning rate (the rate at which the learning process is paced, which is typically defined as $\gamma[k] = \gamma_0 \exp\left( -\frac{k}{k_1} \right)$; $\gamma_0$ is the initial value of learning rate, usually, $\gamma_0 \in [0, 1]$; and $k_1$ is the time constant. $h_{m_j, m_j^*}$ is the neighbourhood function that is defined as a $h_{m_j, m_j^*} = \exp\left( -\frac{D_{m_j, m_j^*}}{2\sigma[k]^2} \right)$ where $D_{m_j, m_j^*}$ is the distance between the neuron $m_j$ and $m_j^*$ and $\sigma[k]$ is the effective width of the topology neighbourhood, which is defined as $\sigma[k] = \sigma_0 \exp\left( -\frac{k}{k_1} \right)$; here, $\sigma_0$ is the initial value of the effective width and again $k_1$ is the time constant. Depending on the representation power that neurons of the $j$ layer provides, the training can be conducted for one of more epochs and by the end of them, each neuron computes its mean quantisation error (MEQ) as:

$$mqe_{m_j} = \frac{1}{\left\| C_{m_j} \right\|} \sum_{x_i \in C_{m_j}} \left\| w^x_{m_j} - x_i \right\| \tag{4}$$

where $C_{m_j}$ is a subset of the dataset represented by neuron $m_j$, i.e., the data points whose BMU is $m_j$. It is worth noting that we define *mqe* in terms of the feedforward weight vectors without considering the feedback weight vector. This definition is justified by the fact that the feedforward weight vectors connect the neuron to the input space, and it is the sole responsibility to represent the datapoints. Following the computation of *mqe*'s of all neurons, the *MQE* of the entire map at level $j$ is computed as the mean of *mqe*'s of all BMU neurons in $j$, i.e.,

$$MQE_J = \frac{1}{\|U\|} \sum_{\forall m_j \in U} mqe_{m_j} \tag{5}$$

where $U$ is the subset of $M$ containing all the BMU's. Once these computations are performed, a decision whether there is a need to add more neurons to the same level or add new layers has to be made. Such a decision is performed by comparing the $MQE_J$ with the *mqe* of its parent, i.e., the neuron in the upper level $j-1$ from which the level is emerged, i.e.,

$$MQE_J < \tau_1 MQE_{m_{j-1}} \tag{6}$$

If the value of Equation (6) is evaluated as false, it means that the current map cannot represent the dataset at the desired level of granularity and, therefore, the process of

horizontal growth must be initiated. This process starts by selecting the neuron with the maximum *mqe* value in layer J, i.e., $e_j = \arg\max_{m_j}\left\{D_{m_j}\right\}_{\forall m_j \in M}$ and the furthest neighbour within its receptive field to $e_j$ in terms of the weight vector, denoted by $d_j$. A new set of neurons is then added between $e_j$ and $d_j$. The new map architecture is then trained and evaluated against the condition given in Equation (6). Once this condition is met, the horizontal growth is terminated, and the vertical growth process begins. The main goal of this process is to determine whether or not each neuron in the current map is placed at the correct level. This determination Is made by comparing the *mqe* of all neurons with the *mqe* of the neuron at level 0, i.e., $mqe_0$ using Equation (7). If a neuron does not meet this condition, it is moved to the next level of the map.

$$mqe_{j,k} < \tau_2 mqe_0; \ \forall j, k \in J, K \tag{7}$$

where $J, K$ denoted the sets of all neurons in the horizontal and vertical levels, whereas $\tau_1$ and $\tau_2$ are the hyperparameters of the model whose values are set to 0.05 and 1.0, respectively.

### 3.3. Grossberg Layer Supervised Layer

The output layer, as defined in the original CPN architecture, is a single layer with one or more artificial neurons, each of which is fully connected to all other neurons in the SOM layer. Although this makes this layer looks like an MLP architecture, it differs significantly from that architecture both in the way by which the weighting connections are updated and, in the strategy used to perform the learning. In this layer, the actual target values (ground truth) are used to perform the learning process, whereas in the traditional MLP network, the magnitude of the deviation of the target value from the predicted value (i.e., the prediction errors) is used instead. Using the actual values not only speeds up the convergence of the model, but also reduces the possibility of trapping into local minima, which typically occurs when the error is too small to be captured by the learning rate. Furthermore, the output of this layer uses the Grossberg learning rule, where the new value of the weights is calculated based on the value of the current weight, the ground truth and the output of the SOM layer, without the need for complicated mathematical operations (i.e., as gradients in the MLP architecture). The main advantage of the Grossberg learning rule, which, besides its low computational cost, has a high level of robustness against data deviations. More specifically, adjusting the weights of the neurons in this layer in accordance with all the fired/triggered SOM neurons facilitates the retention of valuable available information related to various clusters derived from the unsupervised learning strategy in the mapping space of each neuron. It is worth noting that the artificial neurons of the CPN output layer do not contain an activation function, as is the case with their counterparts in the MLP networks. As a result, the CPN output layer avoids the limitations associated with selecting an inappropriate activation function, such as output space constraints, potential bias shifts, and lack of smoothness.

## 4. Results and Discussion

This section is devoted to the results and discussion of this study. First, Section 4.1 describes the dataset used to evaluate the performance of the model proposed in this study. Section 4.2 provides an overview of the performance metrics that were used to quantify the evaluation. Sections 4.3–4.5 provide the evaluation that was conducted to assess the model's learning ability, its evolving process, and comparison with related works, respectively.

### 4.1. Overview of the C-MPASS Dataset

With the aim to assess the validity of the proposed model from different perspectives and under different situation, two datasets are used here: commercial modular aero-propulsion system simulation (C-MPASS) [52]. C-MPASS was created by the National

Aeronautics and Space Administration (NASA) and made publicly available at Ames prognostics data repository. The repository of this dataset consists of 13 text files totalling 42.8 MB for 709 training and 707 engines data, which can be divided into four groups named FD001, FD002, FD003, and FD004. The four files, prefixed by the word "train", contain temporal readings from 21 gas turbine engine's sensors and three operating settings that have significant effects on engine performance. In addition, each record/row in a training file is indexed by a tuple, consisting of engine identification and a cycle number at which the 21 sensory readings are taken. The initial wear and manufacturing variation of each engine are not listed in the files, however, the information of the last cycle of a given engine is the instance at which the engine reaches failure case. This, in turn, makes each file a set of multivariate time series of run-to-failure cycles. Thus, the remaining useful life (RUL) of each unit/engine in the training datasets can be calculated by counting the number of cycles for that unit. The four test files, preceded by the word "test", have an identical structure to the training files, except that the sensory readings are given for a subset of cycles during which the engine is fully functional, so the RUL cannot be calculated directly from these files. Instead, the real RUL values of each engine in the test datasets are given in separate files whose names are prefixed with the word RUL. Table 1 compares these sub-datasets from different perspectives including the number of training and testing trajectories, the number of conditions (e.g., sea level) under which the data were simulated, and, finally, number of fault mode (e.g., HPC degradation, fan degradation).

**Table 1.** Description of the C-MPASS dataset.

| Parameter | FD001 | FD002 | FD003 | FD004 |
|---|---|---|---|---|
| Number of training trajectories | 100 | 260 | 100 | 249 |
| Number of testing trajectories | 100 | 259 | 100 | 248 |
| Number of conditions | 1 | 6 | 1 | 6 |
| Number of fault modes | 1 | 1 | 2 | 2 |

*4.2. Performance Metrics*

Mean absolute error ($MAE$) is one of the performance metrics widely used to measure the overall ability of regression models to make accurate predictions. $MAE$ is defined as the average Manhattan distance between the predictions generated by the model and the corresponding actual values given by the dataset providers. In mathematical notation, $MAE$ can be expressed as follows:

$$MAE = \frac{1}{\sum_{\forall e \in \mathcal{E}} t_e} \sum_{i=1}^{\sum_{\forall e \in \mathcal{E}} t_e} \left\| \hat{\mathcal{R}}_i - \mathcal{R}_i \right\| \tag{8}$$

Recalling that $\sum_{\forall e \in \mathcal{E}} t_e$ represents the total number of monitoring cycles of all assets accommodated in the dataset, $\mathcal{R}_i$ is instances of the actual RUL values, (i.e., $\forall \mathcal{R}_i \in \mathcal{R}$) whereas $\hat{\mathcal{R}}_i; \forall \hat{\mathcal{R}}_i \in \hat{\mathcal{R}}$ are the corresponding predicted values.

While $MAE$ provides valuable insight into the predictive performance of a model, its reliance on the calculation of errors using Manhattan distances makes it less sensitive to effectively accounting for outliers in the dataset. A possible solution to this limitation is to use mean square error (MSE), which replaces absolute differences with squared differences so that larger deviations from the norm are weighted more heavily. However, since the unit used to measure this error is the square of the physical unit, the square root of the MSE, i.e., $RMSE$, is often preferred and formulated as:

$$RMSE = \sqrt{\frac{1}{\sum_{\forall e \in \mathcal{E}} t_e} \sum_{i=1}^{\sum_{\forall e \in \mathcal{E}} t_e} \left\| \hat{\mathcal{R}}_i - \mathcal{R}_i \right\|} \tag{9}$$

However, one of the main limitations of $MAE$ and $RMSE$ is that they are not able to quantify the directions of the errors. The use of absolute values and squared terms in

these two metrics obscures the distinction between whether the errors are due to over-prediction (i.e., when the predicted values are higher than the corresponding actual values) or under-prediction (i.e., when the predicted values are lower than the actual values). In view of this limitation, various performance metrics such as the residual error or the mean bias error have been proposed. However, in the field of RUL, it is crucial to treat over-predictions and under-predictions differently, as over-predictions can lead to higher MRO costs, while under-prediction can have catastrophic consequences. Inspired by this need, some references [25–37] have introduced the scoring index as:

$$Score = \begin{cases} \sum\limits_{i=1}^{\sum_{\forall e \in \mathcal{E}} t_e} e^{-\left(\frac{(\hat{\mathcal{R}} - \mathcal{R}_i)}{13}\right)} - \mathbf{1} \; ; \; \hat{\mathcal{R}} - \mathcal{R}_i < 0 \\ \sum\limits_{i=1}^{\sum_{\forall e \in \mathcal{E}} t_e} e^{\left(\frac{(\hat{\mathcal{R}} - \mathcal{R}_i)}{10}\right)} - \mathbf{1} \; ; \; \hat{\mathcal{R}} - \mathcal{R}_i \geq 0 \end{cases} \tag{10}$$

### 4.3. Assessment of the Model Learnability

The first evaluation of the proposed model focuses on assessing the prediction errors generated by the model during its training phase. For the purpose of this evaluation, four identical models were created as described in Section 3, each trained on a subset of the C-MPASS dataset: FD001 to FD004. During this evaluation, the normalized $MAE$ was recorded at the end of each epoch and plotted as shown in Figure 2.

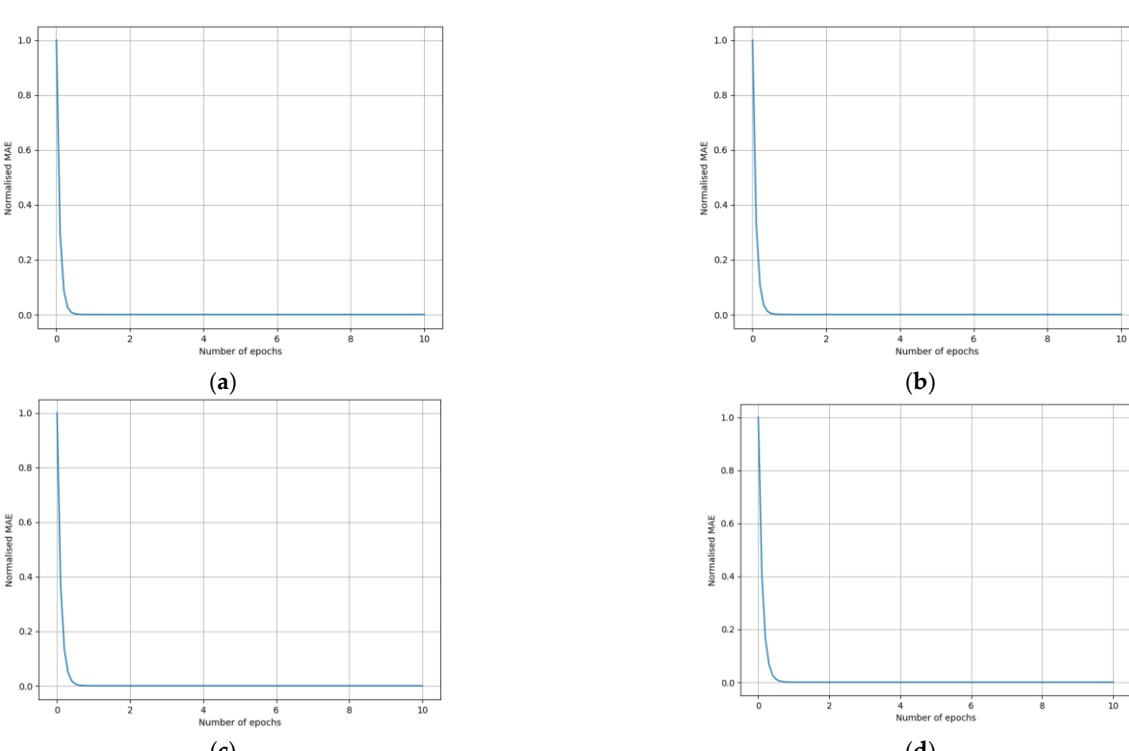

**Figure 2.** $MAE$ vs. number of epochs of the C-MPASS dataset. (**a**) $MAE$ vs. number of epochs of FD001. (**b**) $MAE$ vs. number of epochs of FD002. (**c**) $MAE$ vs. number of epochs of FD003. (**d**) $MAE$ vs. number of epochs of FD004.

The results depicted in Figure 2 show that the normalized $MAE$s of the four models decrease monotonically over time until they settle in quasi-negligible regimes. Furthermore, these results also show that the number of epochs required for the different models to reach these regimes is approximately the same, even though each model was trained on a different subset of data. These results corroborate the ability of the proposed model to make accurate predictions for RUL values under various conditions and fault modes. Indeed, utilising the counter-propagation principles as a vehicle for developing the model

allows it to take advantage of both unsupervised and supervised learning strategies. Thus, with a simple architecture of basic mathematical operations, the model can handle very sophisticated and highly correlated datasets. Unsurprisingly, these results are consistent with findings from the prior articles investigating the performance of counter-propagation networks, e.g., [44–49]. For example, it was reported in [44] that the high coverage speed of CPN is due to the fact that the weight-adjusting process does not include all the connections across the whole network, but only the subset of connections connecting the winning node (best matching unit) of the given training instance and its neighbours. Another important factor for the fast coverage of the CPN is that the training process of the unsupervised subnetwork (recursive growing hierarchical self-organising map (ReGHSOM) in this study) and the supervised subnetwork (Grossberg network) are performed in tandem. This, in turn, means that the computational complexity associated with the training process does not scale up with the depth of the networks. The high resistance of the CPN to noise and outliners is another strong aspect to consider when interpreting the results in Figure 2. In particular, mapping the original dataset into a SOM map, which is known for its ability to preserve topological order, makes it easier to isolate abnormal readings from others. Thus, when you perform supervised learning over the resulting map, the effects of these readings are filtered out.

*4.4. Assessment of the Model Evolving*

In light of the above, a further evaluation of how the proposed model was able to achieve these results is presented here. This evaluation is achieved by presenting the output of ReGHSOM, presented in the form of the dendrogram; the results of these evaluations can be seen in Figure 3.

As can be seen in Figure 3, the ReGHSOM dendrograms for the four subsets of data differ significantly, which highlights the ability of ReGHSOM to construct the SOM architectures that match the characteristics of the dataset at runtime without imposing any constraints or prior assumptions on these architectures. We recall that ReGHSOM is one of the main contributions of this work, proposed to leverage the capabilities of the original counter-propagation network in processing multivariate time series. Simply put, ReGHSOM combines the recursive SOM and the growing hierarchical self-organising map into a new variant where the growth is based on the temporal structure of the dataset rather than the features of the dataset as is the case with the original GHSOM.

As can be seen from Figure 3a, the dendrogram of the sub-dataset FD001 is quite simple compared to the dendrograms of the other sub-datasets. In contrast, the dendrogram of the FD004 sub-dataset contains a much larger number of layers, nodes, and connections between layers. The dendrogram of FD001 shows that the maximum distance between the root (located at the top of the y-axis of the dendrogram; this node corresponds to the first neuron generated when the map is initialised at level 0) and the leaves (at the bottom of the y-axis of the dendrogram, which corresponds to the BMUs and is connected to the supervised layer) is about 18, while the size of this distance is about twice as large in FD004. The difference in the size of the distances highlights the variability (dissimilarity) exhibited by the datapoints of the sub-datasets where the greater the distance, the greater the dissimilarity, and vice versa. Another important observation that emerges from the comparison of the dendrograms of FD001 and FD004 is the number of clades and their types. The dendrogram of FDD4 has many clades, most of which are simplicifolious (the branch of the dendrogram of a single leaf), while the dendrogram of FD001 has fewer clades of tri- or more folious. The increased number of simplicifolious clades in the dendrogram is an indication of the increase in the number of odd observations, i.e., observations that cannot be clustered with other observations because of their unique characteristics. The level at which the clades are generated is another important indicator of how the contents of the datasets are related. The dendrogram of FD001 shows that a considerable number of clades are formulated at the top and middle levels, while FD004 shows that the majority of clades are formulated at the lower level. From these differences in the levels at which the

clades are formed, it can be inferred that most of the datapoints in FD001 have relatively high similarities, while in the case of FD004, these similarities are limited to smaller groups of datapoints. A justification for our discussion can be found by considering the fact that the FD004 sub-dataset was simulated to represent six different conditions with two fault modes: HPC degradation and fan degradation, while the FD001 sub-dataset was generated under a single condition and a single degradation mode. The use of a higher number of conditions and fault modes during the simulation session inevitably leads to a higher variation in the dataset. Further evidence of this can be found in [20], where it is stated that the FD004 sub-dataset can be seen as a general case for the other subsets. An even more interesting observation regarding some engines in FD004 was made by [60], where the authors of this work show that some units alternate between healthy and faulty cases during their lifetime. A look at the dendrograms of FD002 and FD003 shows that they represent intermediate cases between FD001 and FD004.

A further evaluation of the proposed model from the perspective of prediction accuracy is shown in Figure 4. From these figures, it can be seen that the proposed model can determine the RUL for all engines in the four sub-datasets with high accuracy.

### 4.5. Comparison Prediction Accuracy with Related Works

The results presented in Section 4.2 demonstrate the ability of the proposed model to provide exemplary learning curves in front of multiple datasets with different conditions and fault modes. This outstanding ability was further explored in Section 4.3 by demonstrating how the model can dynamically improvise the architectures that fit the characteristics of the given dataset at runtime. In this section, an evaluation of the proposed model from a new perspective is presented. This evaluation was designed to quantify the overall performance exhibited by the model during the training and testing phase. The *RMSE* and *score* metrics of the four models developed to process the subsets of the CMAPSS datasets are measured and then compared to their counterpart readings reported by selected related works. The results of this comparison are presented in Table 2, where the first column contains the reference number, the second column shows the year in which the paper was published, the third column summarises the AI architectures, while the remaining columns show the results ordered by the four sub-datasets. The following abbreviations are used to denote the architectures: LSTM for L\long short-term memory, BiLSTM for bidirectional LSTM, CNN for convolutional neural networks, MLP for multilayer perceptron, EWC for elastic weight consolidation, DRL for deep reinforcement learning, IGRU for involution gated recurrent unit, RGCN for recurrent graph convolutional network, STG for spatial–temporal graph, BLS for fusing broad learning, and TCN for temporal convolutional network.

**Table 2.** Comparison with related works.

| Ref. | Year | Architecture | FD001 | | FD002 | | FD003 | | FD004 | |
|---|---|---|---|---|---|---|---|---|---|---|
| | | | *RMSE* | *Score* | *RMSE* | *Score* | *RMSE* | *Score* | *RMSE* | *Score* |
| [25] | 2016 | CNN | 18.44 | 1286.7 | 30.29 | 1375.0 | 19.81 | 159.62 | 20.15 | 788.64 |
| [26] | 2018 | CNN | 12.61 | 273.7 | 22.36 | 10,412 | 12.64 | 284.1 | 23.31 | 12,466 |
| [28] | 2020 | EWC | 12.56 | 231 | 22.73 | 3366 | 12.10 | 251 | 22.66 | 2840 |
| [22] | 2022 | LSTM and MLP | 7.78 | 100 | 17.64 | 1440 | 8.3 | 104 | 17.63 | 2390 |
| [23] | 2022 | LSTM | 11.35 | 213.65 | 17.78 | 1512.18 | 9.65 | 191.37 | 22.21 | 3285.51 |
| [30] | 2023 | DRL | 12.17 | 208.06 | 16.28 | 1436.81 | 13.08 | 225.50 | 18.87 | 1725.74 |
| [31] | 2023 | IGRU | 12.34 | 238 | 15.59 | 1205 | 13.12 | 292 | 13.25 | 1020 |
| [32] | 2023 | CNN and GRU | 16.29 | 270.78 | 31.46 | 1014.90 | 23.71 | 583.14 | 41.13 | 1722.93 |
| [33] | 2023 | LSTM and CNN | 3.52 | 29.98 | 13.29 | 693.46 | 4.44 | 32.96 | 13.79 | 720.64 |
| [34] | 2023 | RGCN | 11.18 | 173.59 | 16.22 | 1148.16 | 11.52 | 225.03 | 19.11 | 2215.9 |
| [35] | 2023 | STG | 11.62 | 203 | 13.04 | 738 | 11.52 | 198 | 13.62 | 816 |
| [36] | 2022 | BiLSTMA | 13.78 | 255 | 15.94 | 1280 | 14.36 | 438 | 16.96 | 1650 |
| [37] | 2022 | BLS and TCN | 12.08 | 243.0 | 16.87 | 1600 | 11.43 | 244 | 18.12 | 2090 |
| PMTr | 2023 | CP and ReGHSOM | 1.87 | 16.1 | 8.51 | 521.01 | 2.47 | 12.24 | 8.14 | 624.87 |
| PMTs | 2023 | CP and ReGHSOM | 1.57 | 15.5 | 8.24 | 522.31 | 2.35 | 12.50 | 8.78 | 622.45 |

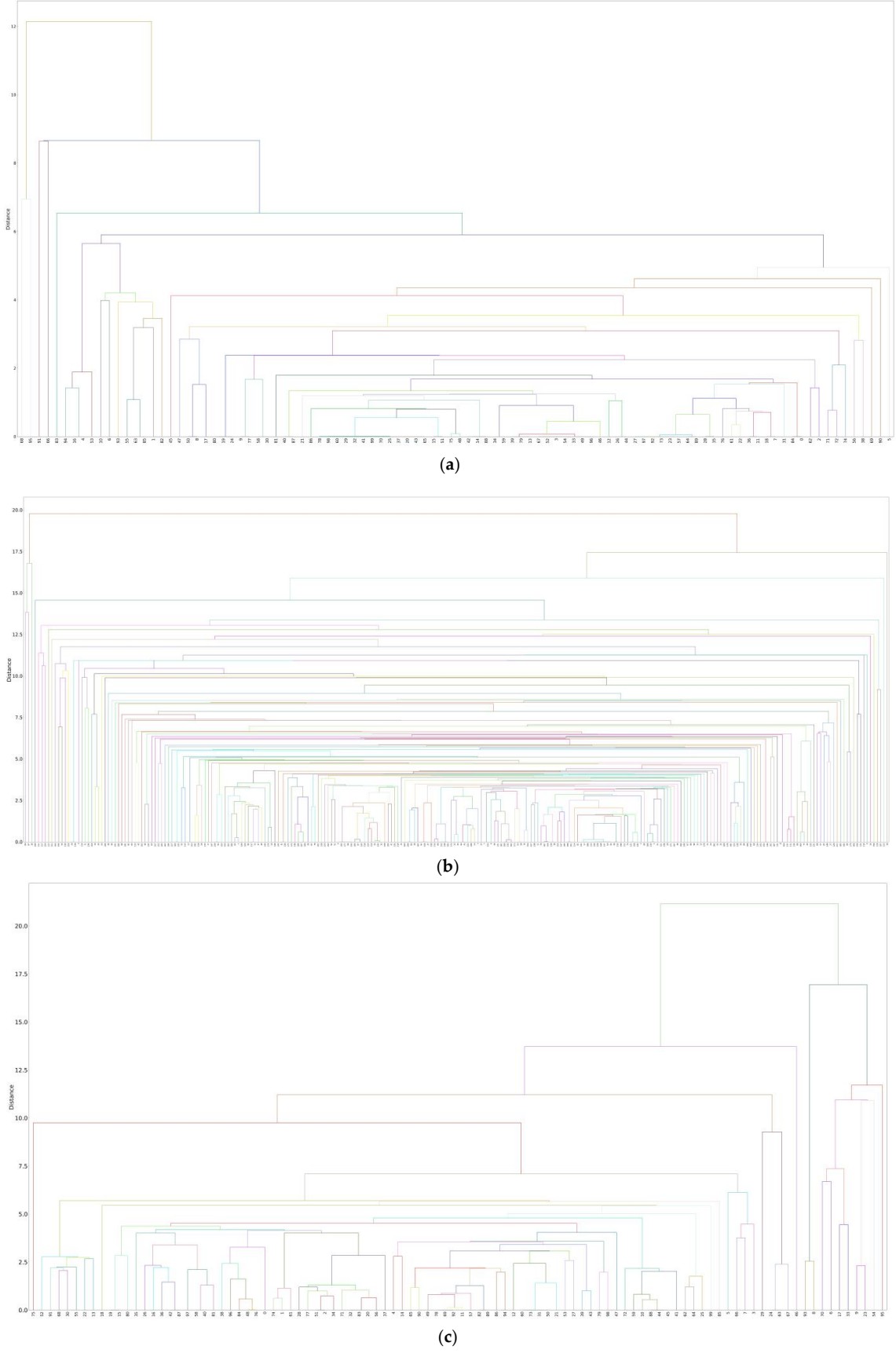

**Figure 3.** *Cont.*

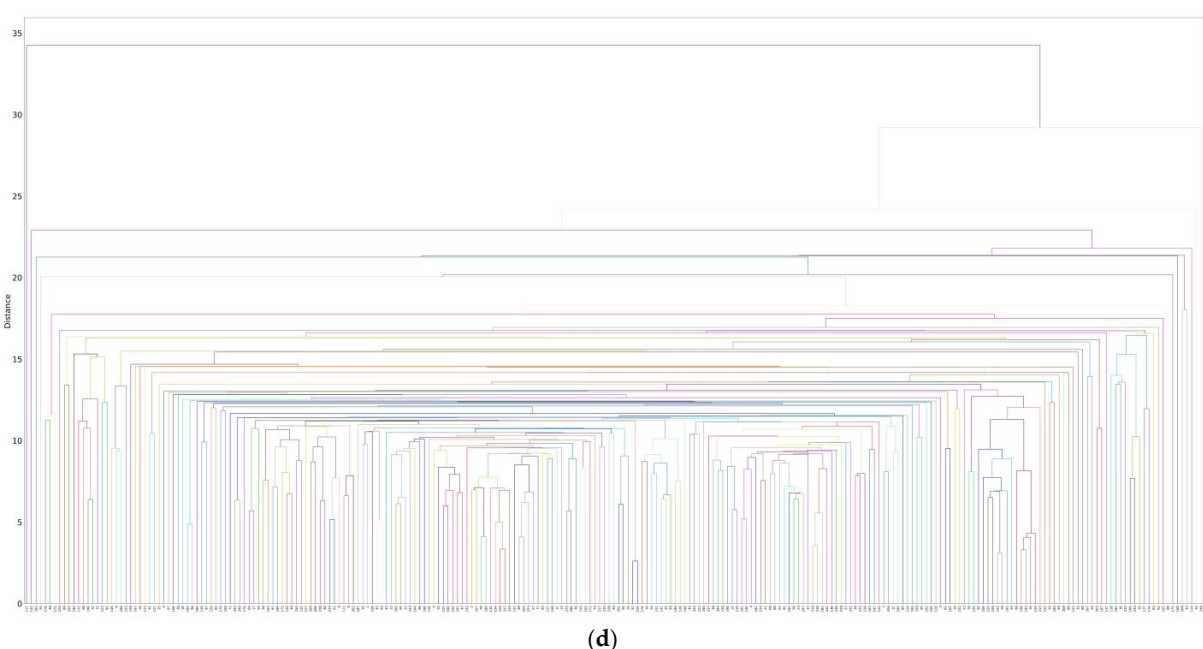

(**d**)

**Figure 3.** Dendrogram of the C-MPASS dataset. (**a**) Dendrogram of FD001. (**b**) Dendrogram of FD002. (**c**) Dendrogram of FD003. (**d**) Dendrogram of FD004.

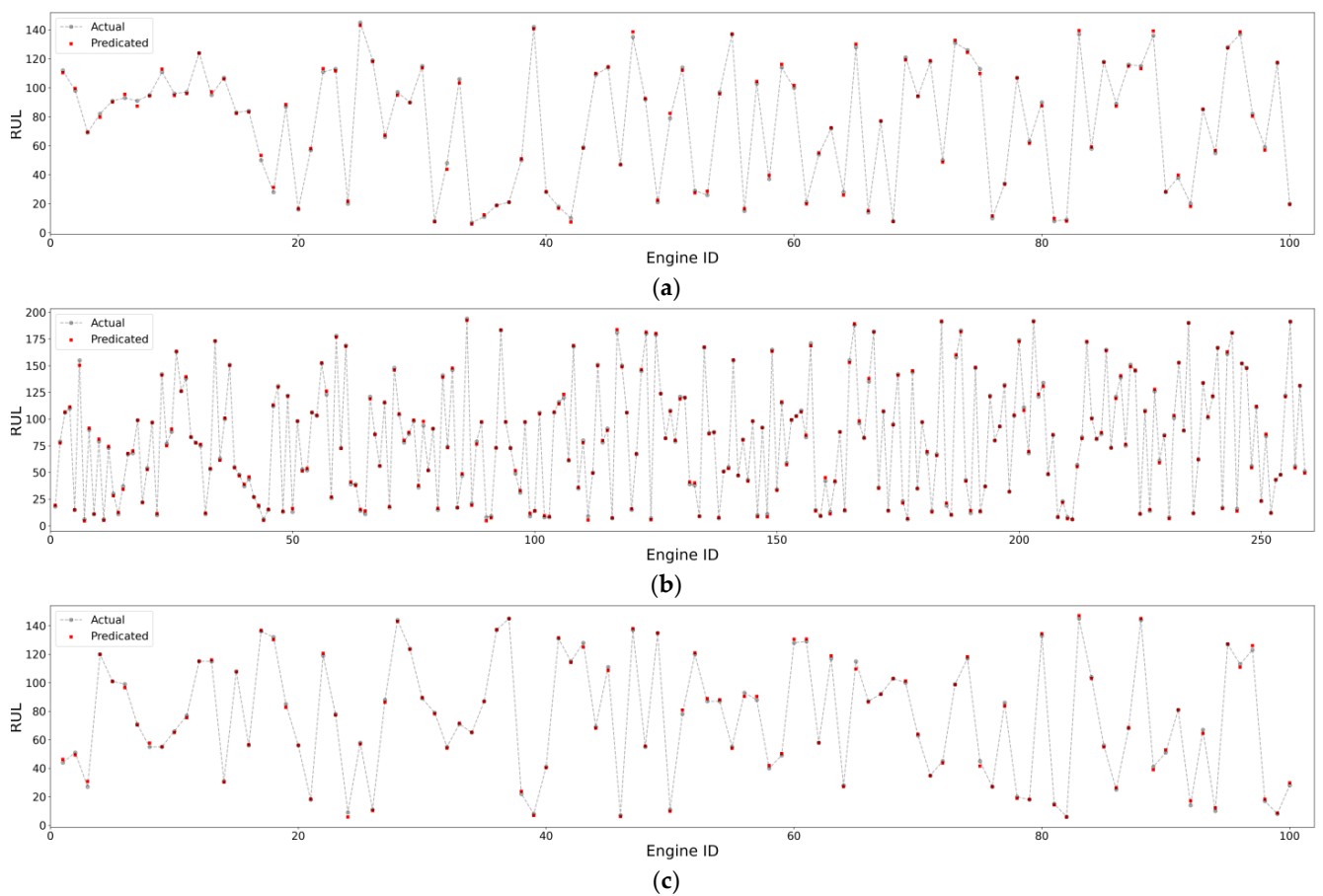

(**a**)

(**b**)

(**c**)

**Figure 4.** *Cont*.

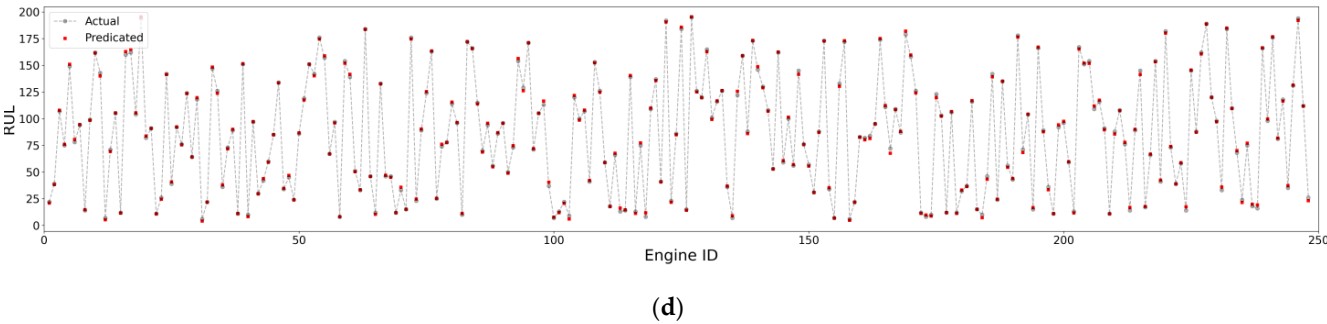

(**d**)

**Figure 4.** Prediction curves for RUL values of C-MPASS dataset. (**a**) FD001. (**b**) FD002. (**c**) FD003. (**d**) FD004.

In general, it can be noted that the proposed model outperforms all other models presented in the works by a competitive rate. This characteristic can be attributed to the fact that the proposed model integrates both unsupervised (using ReGHSOM) and supervised learning strategies (using the Grossberg network) in a way that allows the temporal behaviour embedded in each sub-dataset to be perfectly represented and converted into actual values via a simple architecture and using the same weight update rule. This gives the proposed model and other models based on the counter-propagation network (CPN) approach some strong properties compared to the back propagation (BPN) approach used by most of the comparative works listed in Table 2. Firstly, the fact that the CPN is designed to learn from multiple clusters, each representing the features of a subset of the inputs, allows the CPN to recognise these differences in prediction, which, in turn, improves its accuracy. Indeed, the CPN can be conceptualised as an ensemble learning approach, where the base learners are the BMUs of the resulting SOM map and the aggregation function is the Grossberg network. The BPN, on the other hand, can be seen as a singular learning approach, since the main objective is to reduce the errors generated by the model with respect to all data points collectively. This, in turn, results in the need to apply one or more preprocessing methods to the dataset contaminated with noise and outliers before feeding it to the model. Justification for this claim can be found by reviewing the pre-processing techniques that BPN models have used. The second main advantage of the CPN over the BPN in relation to the scope of this work is the simple architecture and computational budget feasibility that the CPN typically has. As described in Section 3, the CPN does not apply complicated operators (such as gradient computations, matrix multiplications, chain rule, and weight updates, etc.) as the BPN does, nor does it require cascading multiple layers to improve learning performance as the BPN does. Instead, the CPN relies on a simple form of radial function and its architecture is limited to two layers. This, in turn, not only reduces the possibility of the CPN being subject to variations of the model parameters or gradient instabilities, but also eliminates the causes of more difficult problems such as catastrophic forgetting and data/concept drift. A comparison of the values of [28] with [25] shows that the use of EWC, one of the continual learning strategies developed to mitigate the effects of catastrophic forgetting in deep learning, can lead to higher prediction accuracy.

*4.6. Comparison of Computational Complexity with Related Works*

The previous subsection demonstrated the ability of the proposed model to make higher accuracy predictions than some of the state-of-the-art peer works. The focus of this subsection is, therefore, on whether or not this outperformance is at the expense of higher resource utilisation. To this end, several deep learning models with architectures similar to those considered in Section 4.4 were created with particular emphasis on this and evaluated using the following metrics: (1) number of model parameters that count the total number of weights and biases that the model uses to map the given input to the desired output; (2) FLOP counts, which is the total number of FLOating point operations the model performs during its processing for a single instance of the input; (3) the prediction time, which

is the number of CPU seconds required by the model to generate the output corresponding to a given instance of the input in the test phase. The results of this evaluation are summarised in Table 3.

**Table 3.** Computational complexity of the proposed model and other deep learning model.

| Architecture | Number of Parameters | FLOP Count | Prediction Time (CPU sec) |
|---|---|---|---|
| CNN | 626,442 | 792,101 | 39.4 |
| LSTM | 752,412 | 851,025 | 44.5 |
| DRL | 1,154,201 | 781,241 | 53.8 |
| CNN and GRU | 921,410 | 891,021 | 50.3 |
| LSTM and CNN | 981,410 | 951,041 | 53.7 |
| RGCN | 890,124 | 984,024 | 52.1 |
| BiLSTMA | 1,012,410 | 892,120 | 52.9 |
| Proposed (FD001) | 35,100 | 12,411 | 13 |
| Proposed (FD002) | 78,510 | 34,152 | 25 |
| Proposed (FD003) | 45,412 | 25,102 | 20 |
| Proposed (FD004) | 98,012 | 40,241 | 29 |

A quick look at the readings in Table 3 shows that the recurrent graph convolutional network (RGCN) prominently stands out with the highest FLOP count. This can be ascribed to the large number of non-affine tensor transformations that this architecture requires. Specifically, in this architecture, the raw input (i.e., the multivariate time series of sensor readings) is transformed into a weighted adjacency matrix to make it compatible with the shape of the input layer of the graph neural network. This matrix is then meticulously processed by several networks of different dimensionality (e.g., LSTM, graph convolutional networks, and 1D and 2D CNN). The aim of this extensive processing is to extract spatiotemporal features before they are concatenated and passed to another graph neural network from which the RUL values are determined. In terms of the number of parameters, it is evident from Table 3 that the deep reinforcement learning (DRL) model with about 1 M parameters exceeds the other models significantly. This considerable parameter count can be attributed primarily to the nature of the reinforcement learning, which relies on manipulating a set of high-dimensional parametric vectors. This manipulation is carried out according to the trial-and-error principle and aims to derive the optimal policy. Consequently, the parameter space of a DRL encompasses all possible states, all potential actions, all prospective state transitions, and the rewards and punishments associated with each transition. It is also worth noting that the LSTM-based models, as indicated in Table 3, exhibit a higher number of parameters and FLOP counts compared to the CNN models. This characteristic can be explained by considering the architectural differences between these two models. An LSTM cell, which is the building block of an LSTM-based model, uses several types of gates with recurrent weights to capture temporal features at both long and short timescales. Therefore, the number of parameters in an LSTM model increases with the number of LSTM units. In contrast, a typical CNN model depends on the sparse connections and global weight-sharing approach to extract the features from the raw data input. The extracted features are then processed by multiple pooling and nonlinearity layers to reduce the dimension of the processed data. This, in turn, leads to more efficient parameter utilisation.

Interestingly, the results in Table 3 show that the proposed model excels in terms of having the lowest number of parameters and the fewest FLOPS counts compared to the other models. The lowest number of parameters is due to the fact that the proposed model consists of only two layers: ReGHSOM and Grossberg, with no hidden layers. The other model, in contrast, was constructed based on deep learning principles, which primarily focus on assembling an abundant number of neurons across multiple layers and employing non-linear activation functions to leverage the credit assignment path. Another important reason for the lowest number of parameters of the proposed model lies in the learning objects that its layers are designed to perform. Recall that the main goal of the first layer of

the proposed model (i.e., the ReGHSOM layer) is to represent the higher-dimensional raw data in a lower-dimensional space, which, in turn, leads to a reduction in the number of parameters that are passed to the next layer. Deep learning typically works in the opposite direction, as the main goal of the foremost layer is to detect the primitive features of the dataset and then propagate them to the hidden layers, where latent feature detection takes place. This, in turn, leads to a significant increase in the number of parameters as the layer depth grows. Furthermore, the combination of GHSOM and ReSOM in the development of the proposed ReGHSOM algorithm offers significant advantages in reducing the number of parameters. This is because ReGHSOM allows the data points to be hierarchically clustered at runtime based on their granularity levels. This, in turn, can avoid redundancy in the representation of closely spaced data points with different clusters, as can be the case with algorithms with a fixed number of clusters. Continuing the argumentation shows that the second layer of the proposed model has a simple architecture whose input dimensionality is shaped by the number of best matching units (BMUs) generated by the ReGHSOM layer. This, in turn, leads to fewer parameters. In contrast, in a typical deep learning model, the outputs are generated by one or more MLP layers, which are known for their high connection density. The main resonance for using MLP is to aggregate the various features extracted from previous layers in a way that preserves their contextual information. The low FLOP count of the proposed model is primarily due to the fact that in counter-propagation models, the training of the different layers is performed sequentially. This not only obviates the need to use non-linear transformations to update the weighting matrix, but also keeps its size as compact as possible. In addition, the homogeneous recurrent links between the feedforward and feedback layers in the ReGHSOM help to maintaining this compact size. Another notable factor contributing to the low FLOP count is the restriction of weight updates to the connections between the best matching units (BMUs) and their neighbouring nodes. This typically leads to a sparse matrix, and, interestingly, this sparsity increases as more data points are assigned to their respective clusters. This sparseness reduces the computational load and, thus, the total count of FLOPs.

A comparison of the time required for the predictions, as indicated in the last column of Table 3, emphasises the exceptional efficiency of the proposed model. This efficiency can be primarily attributed to the simplicity of the model, which avoids complex mathematical operations. Instead, it relies on simple distance calculations between data points, thus, minimising the computational effort. The model also benefits from a reduced number of parameters and a minimised number of FLOPs. In contrast, deep learning models often require more time due to several factors. These include a higher number of parameters and a larger number of FLOPs as well as the use of complicated mathematical operations in deep learning, such as the calculation of gradients for each parameter and their bidirectional forwarding through several layers as well as the use of sophisticated non-linear activation functions. Pioneering work investigating the asymptotic complexity of a simple multilayer perceptron (MLP) network shows that the time complexity grows at a cubic rate with respect to the number of parameters [61,62]. More recent work [63] shows that the complexity of each layer of a CNN scales with the dimensionality of the input, the kernel size, and the square of the representation dimension.

The results presented in this section, in conjunction with the previous findings, show the superiority of the proposed model from various points of view. This superiority includes aspects such as prediction accuracy, computational efficiency, and prediction time.

## 5. Conclusions

This paper proposed a novel data-driven remaining useful life (RUL) prognostics model based on counter-propagation network (CPN) principles. The CPN approach was chosen because it mitigates some of the drawbacks of the backpropagation approach used in most related work by combining unsupervised and supervised learning strategies over the same architectures. To adapt the CPN to the nature of the RUL dataset, this work introduces the recursive growing hierarchical self-organisation map (ReGHSOM)

as a variant of SOM that can cluster multivariate time series with high correlations and hierarchical dependencies typically found in RUL datasets. Moreover, ReGHSOM is designed to allow this clustering to evolve dynamically at runtime without imposing constraints or prior assumptions on the hypothesis spaces of the architectures. The output of ReGHSOM is fed into Grossberg's supervised learning layers to make the RUL prediction. The comprehensive evaluations conducted in this work have demonstrated the ability of the proposed model to achieve an excellent learning curve and generate the architecture that can thoroughly uncover the latent features of the given dataset. Moreover, the comparison of the performance of the proposed model with related works shows that it is able to achieve an average mean square error of 5.24 and an average score of 293 for the C-MPASS dataset, which is better than most comparable works. Apart from the benefits of applying a high accurate RUL in the aerospace industry, this work paves the way for the application of the counterpropagating algorithm to solve various problems in this industry. Our future work includes the application of other unsupervised and supervised learning techniques within the CPN and the use of the full CPN to create a bidirectional mapping for the RULs.

**Funding:** This research was funded by Taif University.

**Data Availability Statement:** All data used in this study are available from the providers mentioned in the manuscript.

**Acknowledgments:** The researchers would like to thank the Deanship of Scientific Research at Taif University for their support in this work.

**Conflicts of Interest:** No conflicts of interest to report regarding this study.

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
