# Peer review of "A Novel RUL Prognosis Model Based on Counterpropagating Learning Approach"

_aerospace, doi:10.3390/aerospace10110972_

Round 1

Reviewer 1 Report

Comments and Suggestions for Authors

This manuscript proposes a data-driven approach based on Counter Propagation Networks for remaining useful life prediction, which is validated on the C-MPASS dataset. The manuscript also has the following problems:

1. When describing the proposed model in section 3, some figures should be included to aid in understanding large sections of text. Currently, just reading the text is not intuitive enough.

2. Besides the dendrogram and Table 2, the prediction curves for RUL can be added to the results in Section 4.

3. Section 1 and Section 2 have roughly the same structure, with the importance of RUL prediction for the aerospace industry, as well as existing prediction methods and the shortcomings of existing methods. Could these two sections be combined into one and then streamlined more?

4. Table 2 shows that the proposed method outperforms most of the methods in terms of prediction accuracy. So is the complexity of the model more complex or simpler compared to the rest of the methods?

5. Can prediction time also be used as a metric to evaluate the model?

Comments on the Quality of English Language

1 What is 'MRO' in Abstract?

2 Some sentences are too long to understand.

Author Response

Dear Reviwer

Please find our reply in the attached file

Reviewer 2 Report

Comments and Suggestions for Authors

The paper describes a prognostic method for determining the RUL of gas turbine engines. It is a well-presented work in that it gives a thorough literature review. As a work, it steps on existing knowledge to introduce its proposed method in a fully justifiable manner. The method exhibits novelty characteristics. The major drawback of this work to this reviewers' view, is the fact that it lacks an engineering character as it is not clear:
i) Whether this approach could be applied to any engine
ii) Is it referring to the RUL of the engine or that of certain components and how are these defined?
iii) In which way the proposed approach will substitute or aid the current approach where critical engine components are checked with non-destructive methods and given life extensions?
A relevant paragraph in the introductory section addressing such issues will ameliorate the quality of the paper. Moreover, it is repeatedly stated in the manuscript that the developed method performs better than most of the counter approaches. This statement needs to be more precise: Which methods perform better? For which reasons? Why is the proposed method still better than those? Finally a minor tipo error: A lot of reference brackets are empty ([])

Author Response

Dear Reviewer

Please find our reply in the attached file

Round 2

Reviewer 1 Report

Comments and Suggestions for Authors

I would like to thank the author for seriously considering my comments and making the changes. I have no more comments.

Comments on the Quality of English Language

I would like to thank the author for seriously considering my comments and making the changes. I have no more comments.